# Daytime Neuromuscular Electrical Therapy of Tongue Muscles in Improving Snoring in Individuals with Primary Snoring and Mild Obstructive Sleep Apnea

**DOI:** 10.3390/jcm10091883

**Published:** 2021-04-27

**Authors:** Peter M. Baptista, Paula Martínez Ruiz de Apodaca, Marina Carrasco, Secundino Fernandez, Phui Yee Wong, Henry Zhang, Amro Hassaan, Bhik Kotecha

**Affiliations:** 1Departmento de ORL, Clinica Universidad de Navarra, 31008 Pamplona, Spain; sfgonzalez@unav.es; 2Servicio de ORL, Hospital Doctor Peset, 46017 Valencia, Spain; pmruizdeapodaca@gmail.com (P.M.R.d.A.); marinacll@gmail.com (M.C.); 3Queen’s Hospital, Barking, Havering and Redbridge University Hospitals NHS Trust, Rom Valley Way, Romford Essex RM7 0AG, UK; Phuiyee_wong@yahoo.co.uk (P.Y.W.); henry.zhang@nhs.net (H.Z.); amrohasaan@yahoo.com (A.H.); bhikkot@aol.com (B.K.)

**Keywords:** neuromuscular electrical stimulation, tongue, snoring, decibels, sleep, tolerability, mild OSA, sleep disordered breathing

## Abstract

**Study Objectives:** Evaluating daytime neuromuscular electrical training (NMES) of tongue muscles in individuals with Primary Snoring and Mild Obstructive Sleep Apnea (OSA). **Methods:** A multicenter prospective study was undertaken in patients with primary snoring and mild sleep apnea where daytime NMES (eXciteOSA^®^ Signifier Medical Technologies Ltd., London W6 0LG, UK) was used for 20 min once daily for 6 weeks. Change in percentage time spent snoring was analyzed using a two-night sleep study before and after therapy. Participants and their bed partners completed sleep quality questionnaires: Epworth Sleepiness Scale (ESS) and Pittsburgh Sleep Quality Index (PSQI), and the bed partners reported on the nighttime snoring using a Visual Analogue Scale (VAS). **Results:** Of 125 patients recruited, 115 patients completed the trial. Ninety percent of the study population had some reduction in objective snoring with the mean reduction in the study population of 41% (*p* < 0.001). Bed partner-reported snoring reduced significantly by 39% (*p* < 0.001). ESS and total PSQI scores reduced significantly (*p* < 0.001) as well as bed partner PSQI (*p* = 0.017). No serious adverse events were reported. **Conclusions:** Daytime NMES (eXciteOSA^®^) is demonstrated to be effective at reducing objective and subjective snoring. It is associated with effective improvement in patient and bed partner sleep quality and patient daytime somnolence. Both objective and subjective measures demonstrated a consistent improvement. Daytime NMES was well tolerated and had minimal transient side effects.

## 1. Introduction

Sleep Disordered Breathing (SDB) encompasses a spectrum of disorders from Primary Snoring (PS) to Obstructive Sleep Apnea (OSA) and is characterized by the common pathophysiology process of repeated and recurrent collapse of the upper airway during sleep. It has been calculated that nearly one billion adults aged 30 to 69 are estimated to have OSA globally, with the majority (60%) with mild disease (Apnea Hypopnea Index (AHI) ≥ 5 to <15 events per hour) and the remaining 40% with moderate (AHI ≥ 15, but < 30) to severe disease (AHI > 30 events per hour) [1].

Habitual snoring is considered to be a prelude condition to OSA and a key indicator for susceptibility to OSA [2]. Snoring is highly prevalent in the middle-aged population (39 to 69 years) [3,4,5], especially in men and has been associated with reduced sleep and excessive daytime sleepiness even in the absence of apneas [6]. In a large metanalysis and systemic review, snoring was linked to a higher risk of Cardiovascular Disease (CVD) and stroke [7]. When comparing habitual snoring to non-snorers, the pooled adjusted hazard risk for CVD was 1.26 (95% CI 0.98–1.62), for Coronary Heart Disease (CHD) 1.15 (95% CI 1.05–1.27), and stroke 1.26 (95% CI 1.11–1.43).

Similarly, the evidence suggests that mild OSA is associated with an increased risk of stroke [8,9,10] hypertension [11,12,13], a reduced quality of life [14], and a predisposition to early atherosclerosis [15]. Despite these adverse health outcomes, the management of mild OSA remains an area of controversy and debate.

Although the gold standard of treatment for OSA is Continuous Positive Airway Pressure (CPAP), management of mild OSA raises different challenges. Snoring can often be the primary complaint and and yet CPAP is associated with higher intolerance leading to notably poor compliance ranging between 45 to 85% [16]. There are several lifestyle practices associated with snoring (smoking, obesity, drinking, etc.), however, a significant proportion of individuals may snore despite not being associated with these. There is a lack of specific guidelines on treatment for snoring. The American Academy of Sleep Medicine (AASM) and the American Academy of Dental Sleep Medicine (AADSM) recommend that sleep physicians prescribe mandibular advancement devices (MAD), as opposed to no therapy, for adults who request treatment of primary snoring, including people without OSA [17,18].

The most notable change that occurs in the physiology of humans during sleep is the reduction in the tone of the muscles and increased collapsibility of the throat and tongue [19]. It has also been demonstrated that increasing the pharyngeal muscle activity or tone, reduces the collapsibility of the airway [20].

There is a body of literature to state that the use of transcutaneous electrical stimulation in inactive or dysfunctional skeletal muscles significantly improves muscle tone and function recovery [21]. Considering the muscles of the tongue are of the same muscle type, the hypothesis that electrical stimulation of the tongue muscles could lead to a similar effect of improved muscle functionality during sleep would be logical.

A previous proof of concept study using a prototype of eXciteOSA^®^ (Signifier Medical Technologies Ltd., London W6 0LG, UK) in a cohort of 27 patients demonstrated the mean bed partner-reported snoring score reduced significantly by 52% from 6.4 to 3.1 (*p* = 0.001) with over 80% declaring a reduction of >40% in the reported snoring [22].

This clinical trial was designed to expand the clinical findings of a previous study that was published recently that demonstrated a notable improvement in simple snorers and patients with mild OSA [23]. The aim of this multicenter trial was to substantiate both objective and subjective outcomes of the transoral NMES in a larger cohort, evaluate change in sleep study parameters, evaluate tolerability, and observe outcomes related sleep quality and daytime somnolence.

## 2. Materials and Methods

### 2.1. Recruitment

Patients complaining of snoring were recruited from multiple centers including Queen’s Hospital, Romford (Barking, Havering, and Redbridge NHS Trust), UK, Clinica Universidad de Navarra, Pamplona Spain, and Hospital Universitario Dr. Peset, Valencia, Spain. Potential participants received an initial screening phone call. Inclusion criteria included: age greater than 18 years, having a live-in partner to report on snoring (VAS), and a history of more than six months of habitual snoring of >5 days per week. A screening sleep test using the Watch-PAT^®^ 200 device (Itamar Medical Ltd., Caesarea, Israel) was undertaken to confirm AHI < 15. Using a peripheral arterial tonometry (PAT) finger plethysmograph and a standard SpO_2_ probe, the Watch-Pat 200 records the peripheral arterial tonometry (PAT) signal, heart rate, oxyhemoglobin saturation, as well as actigraphy from the built-in actigraph. It also includes a body position sensor and a snore sensor that detects acoustic decibel levels. It uses a very sensitive microphone that responds to snoring and other sounds in the audio range and converts them to a small analog voltage that provides a clear, reliable indication of the presence of these sounds. This microphone is placed at the upper area of the chest.

The participants were invited for a clinical examination which included a clinical airway examination by an ENT surgeon. Inclusion criteria were confirmed and a review of the following exclusion criteria conducted: BMI > 35 kg/m^2^; symptomatic nasal pathology (i.e., septal deviation, nasal polyposis, or chronic rhinosinusitis); tonsil hypertrophy (tonsil size of grade 3 or greater); tongue or lip piercing; pacemaker or implanted medical electrical devices; previous oral surgery for snoring, relevant facial skeletal abnormalities (e.g., syndromic facial deficiencies or severe micrognathia); significant oral disease/conditions. Post clinical examination, patients underwent two consecutive nights of the sleep test using the Watch-PAT^®^ 200 device (Itamar Medical Ltd., Caesarea, Israel) for measuring objective snoring parameters (proportion of sleep while snoring louder than 40 dB (all snoring), 45 dB (moderate snoring), and >50 dB (loud snoring)), AHI, oxygen desaturation index ≥ 4% (ODI), and mean arterial oxygen saturation. If the mean AHI on the sleep studies resulted as <15, patients were consented and included in the trial. A change in the proportion of time snoring >40 dB was considered the primary endpoint. The mean of the two consecutive night studies was used to minimize night-to-night variability.

### 2.2. Procedures

For two weeks prior to commencement of therapy, the participants’ bed partners were asked to rate their partner´s snoring using a Visual Analog Scale (VAS), which ranked the impact of snoring from 1 (no snoring) to 10 (intolerable). Both the participants and their bed partners also completed sleep quality questionnaires, including the Pittsburgh Sleep Quality Index (PSQI) and Epworth Sleepiness Score (ESS) at the end of the two-week period.

Therapy commenced with a face-to-face meeting during which participants were instructed on how to use eXciteOSA^®^ (see Figure 1/Appendix A). The mouthpiece and the control unit are connected by a USB port. This activates the control unit which connects to the smartphone app via Bluetooth connection. The patients were instructed to insert the mouthpiece with two electrodes located above and two electrodes located below the tongue. Bipolar biphasic current was delivered with predetermined stimulation and rest periods, migrating between three low frequencies (0 to 20 Hz). The intensity of therapy (maximum of 15 mV) was controlled by the patient and they were advised to use the maximal tolerable intensity without discomfort [23]. The smartphone app was used to start and stop the device which was set to 20 min therapy. Therapy consisted of a series of pulse bursts with the basic characteristic of 6s burst and 4 s rest. During a typical 20-min therapy period, the pulse frequency changed every 5 min in a defined sequence. After use, the control unit was disengaged from the mouthpiece, cleaned, and stored safely or charged if necessary, using the USB charging cable. Users were instructed to clean the mouthpiece by rinsing with tap water.

Participants were asked to use the device for 20 min once daily for 6 weeks and record their daily assessment of side effects and adverse events. Compliance with therapy was recorded remotely by use of the app. Participant’s bed partners were also asked to record daily subjective assessments of their partner’s snoring using the VAS. Both the participant and their bed partners completed a second set of sleep questionnaires (ESS and PSQI) at the end of the six-week therapy period. In the week following therapy, participants were asked to stop using the device and undertake a second 2-day home sleep study (using Watch-PAT^®^ 200 device (Itamar Medical Ltd., Caesarea, Israel)).

After 6 weeks, participants’ bed partners were asked to continue to record their daily subjective assessment of snoring using VAS for the next two weeks. Following this, a feedback meeting was conducted where data sheets were collected and participants asked for their overall subjective assessment of eXciteOSA.

### 2.3. Statistical Considerations

Data from a previous pilot study of eXciteOSA^®^ [22] was used to estimate the minimum number of people needed to detect a 20% reduction in snoring time with 80% power, based on paired observations of pre- and post-treatment for the proportion of sleeping time snoring of more than 40 dB. The calculated required sample size was 125 snorers with an AHI < 15/h. Based on the outcomes noted in previous literature on objective change in snoring, a 20% reduction was considered a clinically relevant change.

Statistical analyses were performed using a paired sample t-test and/or independent sample t-test for parametric data and non-parametric tests for others. Comparative analysis of the percentage of total sleep time spent snoring at >40 dB was evaluated between the 2-day sleep studies conducted pre- and post-therapy.

Logistic regression was performed to help identify specific correlators of clinical effectiveness by comparing responders (those who experienced a >20% reduction in objective snoring time at >40 dB) with non-responders. Analysis was conducted in four key groups of explanatory variables: demographic variables (age, sex, BMI, smoking, or alcohol history), sleep study parameter variables (AHI, ODI, O_2_ saturation), clinical examination variables (neck collar size, Friedman score, tonsil size, nasal examination), as well as endoscopy examination parameters (simulated snore, Muller and Esmarch maneuver). The effect of each set of the explanatory variables on response was assessed by performing multiple logistic regressions with all explanatory variables included. Data results in patients with mild sleep apnea will be discussed in a future paper.

## 3. Results

Of the 125 people initially recruited, 10 people (8%) failed to complete the trial. One person was excluded from the trial following their initial oral/dental examination due to excessive dental disease needing attention. One person was unable to tolerate the device due to their gagging reflex. The remaining eight potential participants withdrew for unrelated reasons. Two people became aware of their pregnancy during the pre-therapy period, a contraindication of use. Six people withdrew due to changes in their personal circumstances.

### 3.1. Demographics

Of the 115 patients that completed the trial, 73 (63.5%) were men and 42 (36.5%) were women. Sixty-five patients had an AHI between 5 and 15 (mild OSA) and the remaining 50 had an AHI < 5 (primary snorers). The mean age was 46 years, with ages ranging between 24 and 79. Sixty-eight (59.1%) participants regularly consumed alcohol with an average consumption per week of 8.5 units. Thirteen (11.3%) of the participants were smokers. The mean BMI of people who completed the trial was 27.0 (BMI in the range of 20.4 to 34.0). (see Table 1 for further details).

### 3.2. Change in Objective. Snoring-Sleep Study-% of Sleep Time Snoring

The mean reduction in the proportion of time snoring at >40 dB was 41% for the 115 participants cohort (*p* < 0.001, 95% CI: 10.5–15.3%) with 90% of patients demonstrating some reduction in their objective snoring time with the use of the device (Figure 2). Clinically significant changes in the proportion of time snoring were also noted at snoring intensity threshold levels of 45 dB (moderate snoring) and 50 dB (loud snoring) with a demonstrated mean improvement at these thresholds of 52% (*p* < 0.001, 95% CI: 4.74–8.39) and 54% (*p* < 0.001, 95% CI: 2.30–5.06) of the 115 patients, respectively.

The mean AHI for the whole group reduced significantly from 6.85 to 5.03 (*p* < 0.001), and the ODI from 5.68 to 4.33 (*p* < 0.001).

### 3.3. Change in Bed Partner Reported Snoring—Visual Analog Scale

To exclude any transitional change, the comparative analysis of bed partner reported snoring (VAS) was undertaken for the two weeks pre-therapy average as a baseline and compared to the last two weeks of the therapy phase (weeks 5/6). To ascertain if any change observed was sustained post stopping therapy, the average VAS for week 5/6 was compared to the average reported VAS for week 7/8 (i.e., post stopping therapy).

Bed partner-reported VAS scores showed a significant reduction in their perception of their partner’s snoring (mean of 6.1 pretherapy to 3.7 by week 5/6, *p* < 0.001, 95% CI 2.0–2.69). Mean VAS did not change significantly between week 5/6 (VAS score 3.7) and week 7/8 (VAS score 3.8) suggesting that, despite cessation of therapy at week 6, sleep partners did not perceive a relapse in the snoring in the following 2 weeks.

### 3.4. Change in Sleep Quality and Daytime Somnolence Parameters

ESS and PSQI were assessed for both snorers and their bed partners over the pre-therapy and therapy phases. ESS dropped significantly for eXciteOSA^®^ users from 8.4 to 5.8, with a mean change for all 115 participants of 2.6 by week 6 (*p* < 0.01, 95% CI 1.98–3.27). The ESS scores reported by participants’ bed partners were not statistically significant even though a reduction was seen from 6.2 to 5.7 (*p* = 0.22).

PSQI scores reduced significantly for both the participant (7.16 to 5.75, *p* < 0.001, 95% CI 0.89–1.92) and their bed partners (6.87 to 5.94, *p* = 0.02, 95% CI 0.15–1.68). Analysis of PSQI domains for the participants indicated that there was a significant improvement in C1 (sleep quality), C2 (sleep latency), C4 (sleep efficiency), C5 (sleep disturbance), and C7 (daytime dysfunction). No changes were seen in the domains of sleep dysfunction or sleep medication.

### 3.5. Correlating Factors

Logistic regression was performed to help identify specific correlators of clinical effectiveness by comparing responders (those who experienced a >20% reduction in objective snoring time at >40 dB) with non-responders. BMI was found to negatively correlate with potential success in this primary endpoint (odds ratio 0.843, *p* = 0.022) while pre-ESS score correlated positively as a higher ESS was associated with a higher probability of response (odds ratio: 1.134 *p* = 0.024).

No correlations were found with any other factors, including those related to the clinical examination of the neck collar size, tonsil size, Friedman score, and of the nasal valve. Nor were any correlations found to factors determined by static and dynamic naso-endoscopy examination by an ENT surgeon.

### 3.6. Stimulation Intensity

As the device is controlled through a smartphone, several characteristics of the patient´s usage can be remotely observed. Compliance to a prescribed therapy regime of once-daily use was observed to be adhered to at a mean of 83% for the study population. The stimulation intensity levels used by the patients was also assessed. The mean intensity for the therapy progressively increased from a level of 5.9 in week one to 8.9 in week six (intensity range 1 to 15). As an exploratory analysis, the intensities used by the responders were assessed against the non-responders. The data showed no statistical difference in the intensity levels used between individuals that responded or did not respond to the therapy. This would suggest that stimulation intensity was not a determinant of the probability of response with the therapy.

### 3.7. Side Effects and Adverse Events

No serious adverse events were reported during the trial and no adverse events caused any participants to discontinue the trial. Reported side effects were experienced in 17 patients (15%), with the most common side effect of oral pooling of saliva during utilization being found in 12 (10.4%) patients. Additional adverse events included: tongue discomfort, 10 (8.7%); tooth discomfort, 7 (6.1%); tongue tingling 7 (6.1%); filling sensitivity, 4 (3.5%); metallic taste, 3 (2.6%); gagging, 2 (1.7%); tightness in the jaw, 1 (0.9%).

## 4. Discussion

The study sets out to identify changes in objective and subjective indices of snoring in a population of patients with primary snoring and mild OSA (AHI < 15). We have observed improvements in both objective and subjective snoring using validated objective snoring measurements from sleep studies and bedpartner reported VAS in participants with primary snoring and mild OSA. This has been supported by a statistically significant improvement in daytime somnolence (ESS) and sleep quality (PSQI).

Daytime NMES (eXciteOSA^®^) technology was well tolerated with no serious adverse events reported. Unlike CPAP or oral appliances, eXciteOSA ^®^ is a daytime therapy device with a low burden of use for the patient. This makes patient tolerability and the acceptance of the therapy much more feasible, at least during the period of time of the trial.

While the overall AHI for the group (*n* = 115) was not high (6.85 events/h) by design, we still observed a significant reduction in the AHI; however, perhaps not clinically meaningful. Although AHI may not be the most important factor from a clinical perspective, especially in the mild end of the OSA spectrum, it is encouraging to note that the symptomatic indices of daytime somnolence (ESS) and sleep quality (PSQI) showed clinically and statistically relevant changes as well. Data results in patients with mild sleep apnea will be discussed in a future paper.

Several options of treatment are available for the management of snoring though no specific guidelines exist. These options span from (stopping smoking, weight and alcohol reduction, exercise, etc.) to the use of CPAP, MAD, or surgery. Table 2 compiles information as evidenced in 10 studies that specifically describe objective changes in snoring [24,25,26,27,28,29,30,31,32,33]. The table provides a comparison to assess the objective changes in reducing snoring with medicine, surgical palatal stiffening procedures as well as, MAD.

The maximum percentage change shown in Table 2 is 70% for MAD. However, this index merely evaluates the reduction in time in which the highest intensities (5%) of snoring occurs. Two additional studies of MAD indicate a reduction in peak snoring intensity of around 9 to 14%. A 12% drop was also found with the surgical technique of UVPP in time spent snoring above 50 dB.

For eXciteOSA^®^, the mean percentage reduction of snoring above 40 dB was 41%, and this increased progressively for snoring levels above 45 dB (52%) and above 50 dB (54%). Therefore, when compared with the outcomes detailed in Table 2, eXciteOSA^®^ is at least as effective as a surgical technique or MAD devices and shows greater efficacy than comparative standards of care.

Training of the upper breathing musculature to improve OSA is not a new concept. A paper in the BMJ in 2006 showed that the use of the didgeridoo led to improvements in sleep-disordered breathing [34]. A Brazilian group using a defined upper airway muscle exercise regimen reported improvement in sleep-disordered breathing, although the mechanisms behind this are unclear [35]. However, awake state instructional exercises differ notably from awake state NMES. NMES involves the application of an electric current through electrodes placed over targeted muscles to induce muscular contractions and has been shown to activate the muscle to a greater extent than voluntary muscle actions under identical conditions [21]. It has also been used to induce the activity of motor units that are difficult to activate voluntarily. NMES has been shown to result in a change in myofibrillar protein expression, to induce a phenotype shift of fatigue-prone to fatigue-resistant (i.e., fiber Type II to I or IIa changes), with a strengthening of the cytoskeleton [36]. These changes mirror the neuromuscular degeneration associated with SDB [37,38,39]. It is likely that a similar mechanism of action occurs in the oral cavity and leads to the improvement in sleep indices as noted in this study.

In a previous placebo-controlled prospective randomized study of daytime transcutaneous tongue stimulation for OSA (mild to severe OSA) using a different device, it was demonstrated that although the OSA index did not significantly improve, there was a significant reduction in the objective snoring [40]. The number of snoring epochs decreased in the training group (baseline 63.9 ± 23.1 epochs per hour versus 47.5 ± 31.2; *p* < 0.05). A further study using another external daytime neck stimulator to stimulate the tongue for an average of 4 weeks in a mixed patient group of mild to severe OSA, noted a significant drop in Apnea Hypopnea Index (AHI) from 29.2 to 21.2 and a significant reduction in the partner-reported snoring scale from 7.0 to 3.4 on a visual analog scale of 1 to 10 (10 = unbearable snoring) [41].

These previous studies on the use of Daytime Neuromuscular Therapy in reducing snoring in patients with OSA support the outcomes observed in our study, although the specific devices used and therapeutic protocols were different. The method of delivering the therapy differs since the historic publications use a transcutaneous neck stimulation for the genioglossus muscle, whilst the current device uses direct transmucosal stimulation of the tongue to stimulate the intrinsic and extrinsic (Genioglossus) muscles of the tongue. Furthermore, the current study specifically studies the role of this therapy in patients with mild OSA and primary snoring only, as often snoring is the primary symptom in this patient group.

Hypoglossal Nerve Stimulation (HNS) is a surgically implanted nerve stimulator that has become commercially available in recent years. Like CPAP, HNS is designed to overcome the obstructive event during sleep, and not to physiologically change or train the upper airway musculature. HNS is indicated for patients with moderate to severe OSA who have failed CPAP therapy, however, it is also associated with complications such as infections and device malfunction [42]. Although HNS implanted devices have shown promising improvements, cost and appropriate patient selection remains a major limiting factor for widespread adoption and it is currently not indicated for mild OSA [43].

Intolerability of the current treatment options, in particular for early-stage sleep-disordered breathing, emphasizes the need for alternative treatment options. The literature shows that as many as half of all people treated with either MAD or CPAP are likely to experience at least one side effect, intolerability leading to non-adherence is high, and invasive surgery comes with inherent risks that some snorers and their surgeons may not wish to take. This trial of 115 subjects and their bed partners found eXciteOSA^®^ to be an effective and well-tolerated alternative for the treatment of snoring and mild OSA.

### 4.1. Tolerability of eXciteOSA^®^


For the majority of the 115 people who completed the trial, the side effects were minor. The most prevalent effect was the pooling of saliva during therapy, which is expected when an object is placed in the mouth. Similarly, tongue discomfort, metallic taste, and filling sensitivity were predominantly noted in the first few weeks of use and did not prevent any person from completing the course of therapy. Tooth discomfort and filling sensitivity were typically linked to an increase in stimulation intensity and alleviated by repositioning of the mouthpiece or a reduction in the intensity of therapy. Note that eXciteOSA^®^ users were advised to choose an intensity that feels comfortable, reducing the intensity to a more comfortable level, if required.

The American Academy of Sleep Medicine in collaboration with the American Academy of Dental Sleep Medicine undertook a meta-analysis on 11 Randomized Control Trial (RCT) studies, that evaluated in-trial adherence rates with MAD versus CPAP in mild OSA. The metanalysis revealed that the adherence rates for both devices were found to be approximately 4 h of use per night (CPAP adherence was assessed objectively from the download data and MAD adherence was assessed subjectively based on self-reported data) [44]. As eXciteOSA^®^ is a daytime therapy, no direct comparison can be undertaken with night-time adherence. However, compliance to therapy data showed that the trial candidates adhered to the recommended therapy regimen (20 min once a day for 6 weeks) at a mean compliance of 83%.

### 4.2. Limitations

We acknowledge the need for a prospective randomized controlled trial. However, there are inherent challenges in designing a study of a device with an awake stage mode of action on one of the body´s primary sensory organs and presents challenges in creating appropriate blinding. Whilst it would be technically possible to provide users with a deactivated device that does not produce an electrical signal, it may be relatively easy for users to deduce if they are receiving sham or active therapy. Randomizing and conducting comparative analysis against MAD might be a more appropriate future trial design. Another limitation of this study was the limited duration of follow-up, a common issue with many early trials of emerging technologies. The official research statement by the American Thoracic society identified many gaps in the current literature related to mild OSA therapy options and makes recommendations on the research priorities and need for further studies. Further studies are planned to capture long-term outcomes and follow guidance set by the task force [45].

## 5. Conclusions

Daytime NMES (eXciteOSA^®^) is demonstrated to be effective at reducing both objective time spent snoring and subjective bed partner reported snoring. This was associated with effective improvement in patient and bed partner sleep quality and patient daytime somnolence. Daytime NMES was well tolerated and had minimal transient side effects.

## Figures and Tables

**Figure 1 jcm-10-01883-f001:**
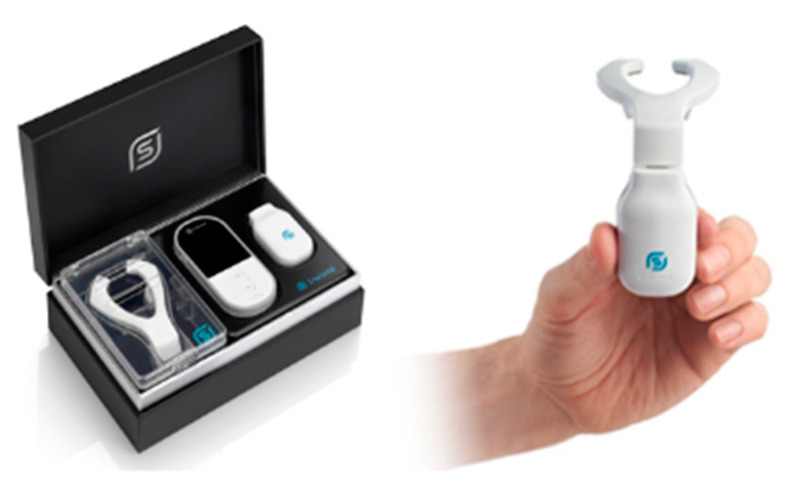
Components of the eXciteOSA^®^. The mouthpiece, USB cable, and control unit comes in a box. After assembling the mouthpiece to the control unit and connecting to the app, the device is placed inside the mouth to begin therapy. Please see the video for more information (Appendix A).

**Figure 2 jcm-10-01883-f002:**
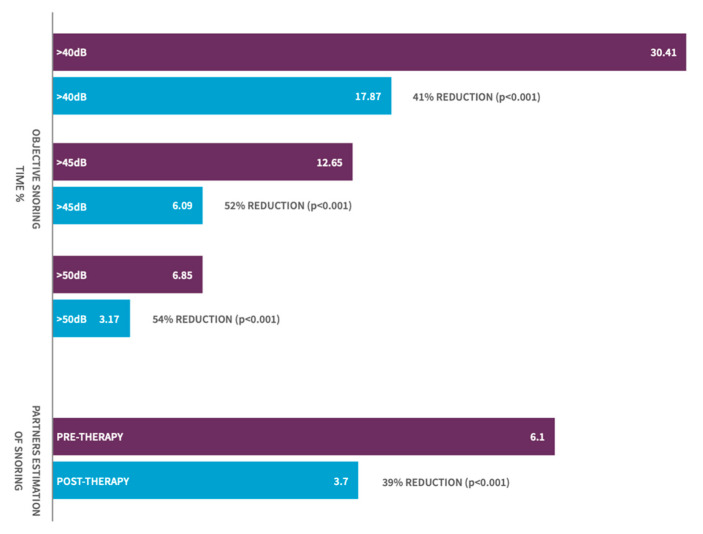
Graph of objective snoring improvement and partner estimation of snoring in patients pre- and post-therapy.

**Table 1 jcm-10-01883-t001:** Demographics.

Descriptive Statistics					
	N	Minimum	Maximum	Mean	Std. Deviation
Age at time of trial	115	24.00	79.00	45.71	13.52
BMI	115	20.40	34.00	27.02	3.26
Pre-AHI	115	0.20	15.00	6.47	4.44
Pre-Patient ESS	115	0.00	22.00	8.06	4.59
Neck Collar (inch)	115	12.50	19.00	15.23	1.49

**Table 2 jcm-10-01883-t002:** Objective reduction in snoring found for devices, medicines, and surgical techniques.

Intervention Type	Intervention	No. People	Measure	Pre	Post	Significance	% Change	Reference
Device	MAD (People with OSA)	22	Mean Peak Snoring Intensity (dB)	71.6 dB	62 dB	Statistical Significance	−14%	Fransson et al. [24]
Device	MAD (People with snoring problems)	13	Mean Peak Snoring Intensity (dB)	63.3 dB	57.5 dB	Statistical Significance	−9%	Fransson et al. [24]
Device	MAD	11/35 *	L5–L95 **	240 (mV)	75 (mV)	Statistical Significance	−70%	Smith and Battagel [25]
Medicine	Budesonide Nasal Drops	24	Mean Snoring Intensity (dB)	61.2 dB	60.1 dB	Not Declared	−2%	Koutsourelakis et al. [26]
Medicine	Injection Snoreplasty (3% Sotradecol into soft palate)	17	Average Loudness (dB)	13 dB	7 dB	Not Significant	−46%	Brietzke and Mair [27]
Surgical	Coblation, Radiofrequency of Soft Palate	21	SNAP Snoring Loudness	12 dB	8 dB	Statistical Significance	−33%	Johnson et al. [28]
Surgical	Laser-Assisted uvulopalatoplasty (LAUP)	27	Average Loudness (dB)	12.7 dB	8.7 dB	Statistical Significance	−32%	Walker et al. [29]
Surgical	Radiofrequency Soft Palate Tissue Reduction	20	Snoring Intensity	60.2 dB	64.9 dB	Statistical Significance	8%	Hukins et al. [30]
Surgical	Somnoplasty (Radiofrequency of soft palate)	10	% Time Loud Snoring	10.62%	8.03%	Not Declared	−24%	Cartwright et al. [31]
Surgical	Somnoplasty (Radiofrequency of soft palate)	10	% Time Spent Snoring	3/7 (42%) people showed improvement in duration of snoring of 30%, 38%, and 48%.				Sandhu et. al. [32]
Surgical	Uvulopalatopharyngoplasty (UVPP)	32	L, level above which 5% of the sound occurs	41 dB	38.8 dB	Statistical Significance	−12%	Pritchard et al. [33]

* A total of 11/35 people were only objectively assessed; ** L5 level is the sound pressure level exceeded 5% of the time in the test period and represents the highest noise levels; L95 level is the noise level exceeded 95% of the test period and represents background noise.

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
