# Peer review of "Daytime Neuromuscular Electrical Therapy of Tongue Muscles in Improving Snoring in Individuals with Primary Snoring and Mild Obstructive Sleep Apnea"

_jcm, 2021, doi:10.3390/jcm10091883_

Round 1
Reviewer 1 Report
The treatment of snoring and mild OSA remains problematic. Although oral appliances and CPAP are effective, they are expensive and CPAP is poorly tolerated, particularly in this group of patients who tend to be less concerned about their symptoms. Many primary snorers only present for treatment on the urging of their bed partners, so therapy adherence can be poor. Hence an easily tolerated therapy for primary snoring/mild OSA would be good. However a distinction needs to be made between primary snoring and mild OSA - even for mild OSA there are consequent morbidities in terms of mood, daytime sleepiness, quality of life, alertness and likely cognitive function.
This paper presents data on a new device, which stimulates the tongue during the day. It would be helpful to have more information in the methods section about how this device is used - whether it starts at a low amplitude and is gradually increased, whether this is patient-directed for comfort/tolerance or for effect, or whether there is a set protocol for up-titration.
Snoring is the primary outcome in this paper, measured objectively and subjectively, including by participants' partners. The objective measurement of snoring by measuring loudness is often problematic and may be inaccurate. Subjective assessment of snoring by bed partner is often inaccurate and unreliable. However this is an important outcome for many patients.
Specific concerns:
"The most notable change that occurs in the physiology of humans during sleep is the reduction in the tone of the muscles and increased collapsibility of the throat and tongue” - actually it is a reduction in conscious state.
“The mean AHI for the whole group reduced significantly from 6.85 to 5.03 (p<0.001), and the ODI from 5.68 to 4.33 (p<0.001).” - although this seems to be statistically significant, probably not a clinically meaningful change.
Logistic regression - predictors of response. Response defined as 20% reduction in objective snoring time at >40dB.
Was device usage and stimulation intensity level included in this assessment?? ie were those who used the device at higher stimulation levels more likely to be responders?
Table Two - formatting is problematic and not easily read or understood
The main concern with this study is the lack of a control group for what are predominantly subjective outcomes. However if this is adequately discussed and acknowledged in the Conclusions, this study should be a useful contribution to the literature. The Discussion and Conclusions should be revised to more accurately reflect the data, including limitations, and focus on the primary outcomes
Reviewer 2 Report
The manuscript is well written in scientific way and with fluent English language. The aim of the article is to find out, can use of neuromuscular electrical training (NMS) of tongue muscles with eXciteOSA device reduce nighttime snoring. You have analyzed sleep parameters objectively by cardiorespiratory polygraphy studies and ‘subjectively’ by conducting sleep quality questionnaires to bed partners. The results have been handled and presented clearly and conclusion include nice review existing methods used upper breathing musculature training to improve sleep disordered breathing. The CPAP device do not suit for all patients and its adherence rate varies a lot. It is estimated that approximately only 4 hours of use per night is reached. So new, easier methods are welcomed for reduction of snoring.
I have few concerns, which need clarification and to be corrected:
Introduction page 2line 58
AHI classification goes normally: Normal AHI <5,Mild 5<= AHI < 15, Moderate disease 15 <=AHI < 30 and severe AHI >= 30
Materials and Methods, page 3 line 93:
You do not describe at all how snoring was measured. What other sensors are included to overnight polygraphy or polysomnography recording?
How the snoring loudness is converted to dB values?
Results, page 5 line 174:
Table 1. Demographics
Check the number of decimals what you are using in your values. Are your measured values so exact that they should report with four decimals? Then those Alcohol and Smoking values are not expressed very clearly, try to put them into the same page.
page 6 line 185
AHI values are integer values, you could not have AHI = 5.03 , it consists number of hypopnea and apnea events. ESS values are integer too.
Figure 2, page 8 line 249
Some value is missing, at least >40 dB objective snoring. What is the units of these values? and p values are good to put in this picture too.
This subjective snoring term is a little misleading. It is not patient’s own subjective estimation or is it? I think that this is bed partner’s estimation, VAS score, some kind of external estimation?
Table 2. is presented twice and formatting is not good (difficult to read).
